# Concept Bottleneck Model with Zero Performance Loss

Zhenzhen Wang[1,2], Aleksander S. Popel[1], Jeremias Sulam[1,2]

[1]Department of Biomedical Engineering, Johns Hopkins University
[2]Mathematical Institute for Data Science, Johns Hopkins University
zwang218@jhu.edu, apopel@jhu.com, jsulam@jhu.edu

Interpreting machine learning models with high-level, human-understandable *concepts* has gained increasing importance. The concept bottleneck model (CBM) is a popular approach for providing such explanations but typically sacrifices some prediction power compared with standard black-box models. In this work, we propose an approach to turn an off-the-shelf black-box model into a CBM without changing its predictions or compromising prediction power. Through an invertible mapping from the model's latent space to a concept space, predictions are decomposed into a linear combination of concepts. This provides concept-based explanations for the complex model and allows us to intervene in its predictions manually. Experiments across benchmarks demonstrate that CBM-zero provides comparable explainability and better accuracy than other CBM methods.

## 1. Introduction

As artificial intelligence (AI) demonstrates remarkable success in diverse domains, concerns regarding interpretability [1, 2], fairness [3, 4], and privacy [5, 6] are also gaining increasing attention. While the complexity of deep learning models enables modeling complex patterns, it also makes the decision-making process opaque. This "black-box" aspect of modern AI systems raises concerns about deploying these models in high-stakes scenarios, and there is a growing demand for more transparent AI systems [7, 8].

Numerous efforts have been made to enhance the interpretability of deep learning models, many of them generating saliency-map style explanations using gradient-based analysis [9, 10], game-theory approaches [11, 12], decomposition-based methods [13, 14], and more. These saliency maps broadly highlight important features of the input, which provides valuable insights into *where*, or to what features, the model attributes some notion of importance. However, saliency maps might not always be sufficient, specifically when the prediction is based on global attributes, such as color, texture, or overall morphology, rather than specific input dimensions [15]. This limitation is particularly evident in challenging tasks like clinical diagnosis, where localizing a particular subregion in medical images may not fully represent important features, and high-level, domain-relevant explanations are needed for more meaningful explanations [16, 17].

Concept-based explanations provide a compelling alternative by explaining classification models with high-level, human-understandable attributes such as color, shape, texture, and objects [15]. The *concept bottleneck model* (CBM) [18, 19] is one such method that consists of two interconnected predictors: a first concept predictor that predicts the presence of specific abstract concepts in some embedded representation of the input, and a subsequent (linear) model that outputs the probability of the class given the presence of such concepts. Since this model explicitly constructs predictions that rely on the presence of concepts, this layer is referred to as a *concept bottleneck*. A crucial challenge of these CBM-based methods is the loss of predictive power that comes with using a surrogate (linear, and interpretable) model in lieu of a standard and more complex alternative. Although numerous efforts have been made to alleviate this issue, such as constructing very complex and large concept banks [20–25], performance drops still exist, especially in complex tasks.

In this work, we propose CBM-zero, a methodology that explains an off-the-shelf, standard black-box model by converting it to a concept bottleneck model. As Figure 1 illustrates, CBM-zero extracts

Second Conference on Parsimony and Learning (CPAL 2025).

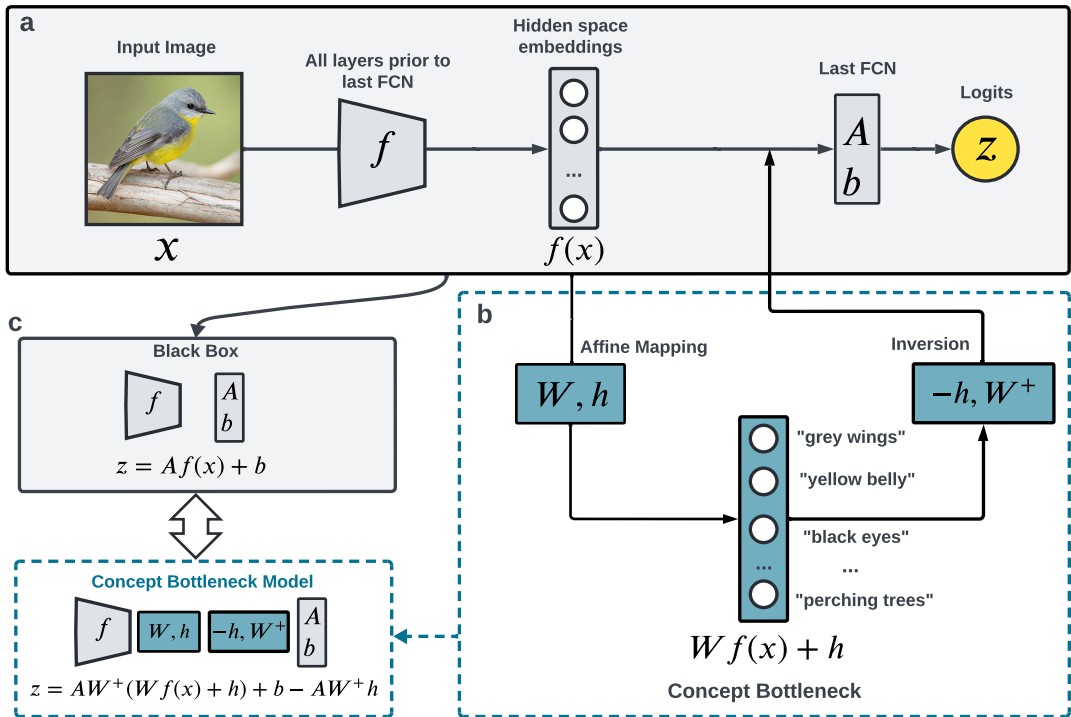

Figure 1: The high-level idea of CBM-zero (**a**) A black-box model. (**b**) Construction of a concept bottleneck through invertible affine mapping from black-box model's hidden space to concept space. (**c**) The black-box model can be reformulated as a CBM without altering predictions.

representations computed just before its *final* layer and finds an affine mapping from this latent space to a concept space derived from human annotations or Contrastive Language-Image Pre-Training (CLIP) models [26]. Importantly, we impose an *invertibility* constraint on the affine map to ensure the original black-box model's prediction is not changed. In this way, We can explain that black-box model in a post-hoc manner without altering it, retaining its performance. Furthermore, like other CBMs, our method allows human interventions. By inspecting and manually fixing incorrect concept attributions, one can modify incorrect predictions. We quantitatively evaluate the accuracy and explanation quality of our method on several image classification benchmarks. Compared with other CBM-based counterparts, our method consistently achieves the best accuracy while offering comparable or better interpretations.

## 2. Related works

As alluded to above, our work is most closely related to CBM-based methods. More broadly, it is also related to conceptual explanation techniques, given that our method aims to explain black-box models with concepts. We review these prior works to put our contribution in context.

**Concept bottleneck model (CBM)**  The initial idea of CBM [18, 19] for classification relied on first predicting concepts and then using these concept scores to predict a class, forming a concept bottleneck. Dense image-wise concept annotations are needed to train the first predictor. Post-hoc CBM (PCBM) [27] proposed learning concept activation vector (CAVs) in feature space as a concept bank [28] and projecting the image embeddings onto these CAVs to produce concept scores. Moreover, in cases where image-wise concept annotations are unavailable, they propose obtaining concept bottleneck by aligning images with concepts through language-vision models such as CLIP. Language in a Bottle (LaBo) [20] and label-free CBM [21] follow a similar idea and further boost the accuracy by collecting concepts from a large language model. More recently, increasingly sophis-

ticated methods have been dedicated to improving the concept bank quality in completeness [22] and flexibility [23, 25]. In all cases, given that the final predictors are based on a linear model based on predicted concepts, all methods share the crucial limitation of performance drops compared to original black-box models. While numerous efforts have been made to minimize this performance gap, some prediction power is always lost in challenging cases. Our method, on the other hand, does not alter the classification function of the original predictor, guaranteeing zero performance drop by design.

**Other conceptual explanations** More broadly, post-hoc methods find alternative ways of explaining an off-the-shelf complex predictor, and typically need image-wise concept annotations in the training set or an auxiliary set. T-CAV [28] is a popular approach that learns a linear classifier on the feature space of a complex model to distinguish samples with and without certain concepts, as parameterized by CAVs. The importance of the concept is then given by directional derivatives of prediction to CAVs. Several extensions generalize and enhance T-CAV. For instance, Automatic Concept-based Explanations (ACE) [29] considers super-pixels in images as concepts and discovers them automatically. ConceptSHAP [30] defines completeness scores for CAVs, and uses Shapley values to quantify the individual importance. Concept activation region (CAR) [31] relaxes the linear separability assumption and uses a region instead of a vector in latent space to represent a concept. Spatial CAV [32] attributes CAVs to relevant spatial regions, while Text2Concept [33] derives CAVs from texts. Casual Concept Effect (CaCE) [34], on the other hand, assesses the causal effect of concepts by generating counterfactual samples. More recently, [35] uses conditional independence of concepts and sequential kernelized testing [36] to assess concept importance. The vast majority of these methods need costly concept annotations to define which concepts to use to probe the latent space of the original, complex model.

## 3. Methods

In this section, we detail the problem formulation, describe our methods, and briefly introduce the related methods to be used in the experimental section.

### 3.1. Problem formulation

Consider a deep learning, black-box model that predicts a label[1] $y \in \mathbb{R}$ from an input image $x \in \mathbb{R}^n$. Most deep neural networks consist of multiple stacked layers and end with a fully connected layer (FCN). Let $f : \mathbb{R}^n \to \mathbb{R}^d$ represent all layers prior to the last FCN. Without loss of generality, we consider a $K$-class image classification task. The weights and bias of the final FCN are denoted as $A \in \mathbb{R}^{K \times d}$ and $b \in \mathbb{R}^K$, respectively. The black-box model and its prediction $\hat{y}$ are then given by

$$z = Af(x) + b, \ \ \hat{y} = \arg\max_i z_i, \tag{1}$$

where $z \in \mathbb{R}^K$ denotes the logits for $K$ classes, and the predicted label $\hat{y}$ is the index of largest logit. The goal is to explain this black-box model with human-understandable *concepts* (e.g., "red", "beak", "stripes"). Like other CBM-based methods, we attempt to solve this problem by defining a *concept bank* as a collection of $M$ concepts, denoted as $S = \{s_1, s_2, ..., s_M\}$. We seek to construct a CBM by mapping the codomain of $f(x)$ to an interpretable concept space, and then invert the mapping to preserve the original predictions unchanged.

### 3.2. Constructing a zero-performance drop CBM

In this section, we will show that any black-box model with the form of Eq. (1) can be converted to a CBM without changing the predictions and completely preserving the accuracy. Figure 2 shows an illustration of our method. The core challenge is that the feature embedding $f(x)$ in Eq. (1)

---

[1]Even though we will study classification problems, we will consider labels in $\mathbb{R}$ as we model the unnormalized logits of the model, which approximate the conditional probability of a label given the input.

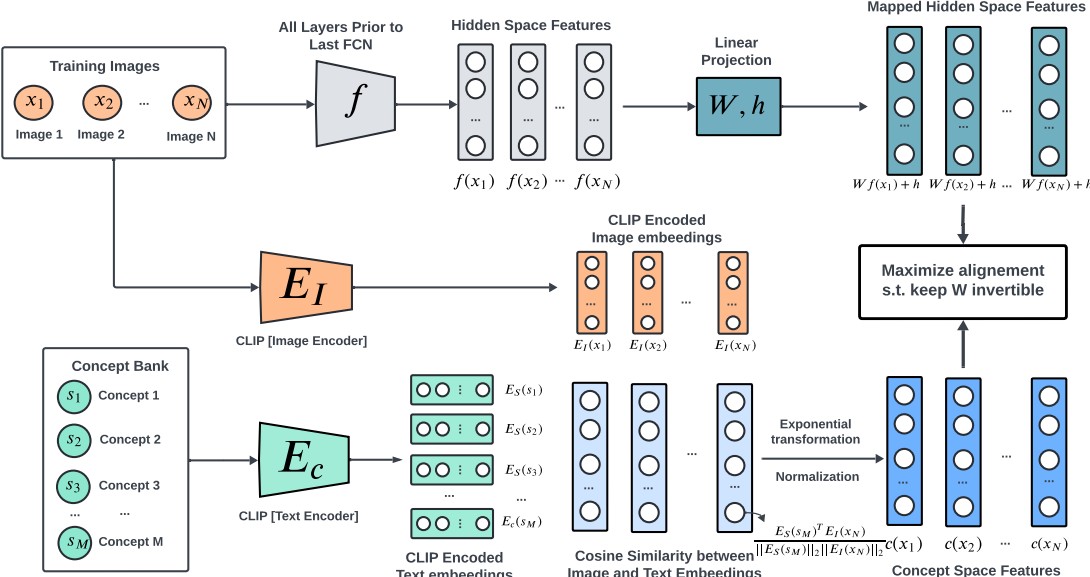

Figure 2: **An overview of the method.** CLIP model is used to estimate concepts' correlation with images. Exponential transformation and normalization are applied to emphasize highly correlated concepts. An invertible affine mapping is learned to map features from a black-box model's hidden space to the concept space.

resides in an abstract space that encodes useful information but is not semantically meaningful to humans. Thus, we aim to relate this hidden space with another one that is interpretable to humans by construction.

To do this, one needs a handle on what concepts are present in each image. In some cases, images are annotated with concept labels, such as in the CUB dataset [37]. However, such dense concept annotations are not always available. Thus, we use the CLIP model [26] to estimate the presence of concepts in images, without requiring external images-wise concept annotations. CLIP trains an image encoder $E_I : \mathbb{R}^n \to \mathbb{R}^l$ and a text encoder $E_S : \mathcal{S} \to \mathbb{R}^l$ (where $\mathcal{S}$ is the space of text/token sequences) jointly via contrastive learning, allowing image and text embeddings to live in a shared space. Their cosine similarity is defined as the CLIP score:

$$\cos(x, s_i) = \frac{E_I(x)^T E_S(s_i)}{\|E_I(x)\|_2 \|E_S(s_i)\|_2}. \tag{2}$$

For an image $x$, the CLIP scores for $M$ concepts yields an $M$-dimensional vector, $[\cos(x, s_1), \cos(x, s_2), \ldots, \cos(x, s_M)]^T$. To address the limited discriminative power of CLIP [38], we apply an exponential transformation to emphasize concepts with stronger correlations, followed by normalization. Specifically, the concept features $c(x)$ is defined as $c(x) = \left[ \frac{\cos^t(x, s_1) - \mu_1}{\sigma_1}, \ldots, \frac{\cos^t(x, s_M) - \mu_M}{\sigma_M} \right]^T$, where $\mu_i$ and $\sigma_i$ are the mean and standard deviation of $\cos^t(x, s_i)$ over the inputs, estimated from training samples. We empirically set $t = 5$ to emphasize concepts with higher CLIP scores, and discuss other choices of $t$ and their sensitivities in Appendix A.3.

We define an interpretable concept space so that $c(x) \in \mathcal{C} \subset \mathbb{R}^M$, and a latent space $\mathcal{H} \subset \mathbb{R}^d$ so that $f(x) \in \mathcal{H}$. We seek an affine projection, $(W \in \mathbb{R}^{M \times d}, h \in \mathbb{R}^M) : \mathcal{H} \to \mathcal{C}$ to map from the hidden space to the concept space. Importantly, we impose the constraint that $\text{rank}(W) = d$, and therefore $M \geq d$, to ensure that the left pseudo-inverse of $W$, defined by $W^+ = (W^T W)^{-1} W^T$, exists. As a result, the mapping from the latent to the concept space is invertible, allowing us to preserve the output of the original model unaltered.

---
**Algorithm 1** Make Matrix Full Rank
---
1: **Input:** A rank-deficient matrix $W \in \mathbb{R}^{M \times d}(M \geq d)$, perturbation scale $\epsilon$
2: **Output:** A full-rank matrix $W'$
3: Perform singular value decomposition (SVD) of $W$: $W = U\Sigma V^T$
4: **for** each singular value $\sigma_i$ in $\Sigma$ **do**
5:    **if** $\sigma_i = 0$ **then**
6:       Sample $r \sim U(0, 1)$
7:       Set $\sigma_i' = \epsilon r$ (small perturbation)
8:    **else**
9:       Set $\sigma_i' = \sigma$ (keep original singular value)
10:    **end if**
11: **end for**
12: Set $\Sigma' = \text{diag}(\sigma_1', \ldots, \sigma_d')$
13: Reconstruct the matrix $W' = U\Sigma'V^T$
14: **Return** $W'$
---

To learn $W$ and $h$, we propose to solve the following optimization problem under the rank constraint:

$$(W_\lambda, h_\lambda) = \underset{W,h}{\arg\min} \underset{x \sim \mathcal{D}}{\mathbb{E}} \|Wf(x) + h - c(x)\|_2^2 + \lambda R(W) \quad \text{s.t.} \quad \text{rank}(W) = d \tag{3}$$

where $\mathcal{D}$ is the training data distribution, $R(W)$ is a regularization term, and $\lambda$ controls regularization strength. We use elastic net regularization on $AW^+$ to encourage sparsity, i.e., so that each class employs a small number of concepts, facilitating interpretability. To be more specific,

$$R(W) = \alpha\|AW^+\|_1 + (1 - \alpha)\|AW^+\|_F^2, \tag{4}$$

where $\|\cdot\|_F$ is the Frobenius norm, $\|AW^+\|_1 = \sum_i \sum_j |(AW^+)_{i,j}|$ is element-wise $\ell_1$ norm, and $\alpha$ controls this trade-off, which we empirically set as 0.5.

The concept bank $S$ is a pre-defined, task-specific set containing concepts relevant to the prediction task of interest. As the black-box model is not initially trained with these concepts, some concepts can be not used or *detectable* by the black-box model. We measure the detectability of $s_i$ by calculating the Pearson correlation coefficient [39] between the $i$th entry of $W_0f(x) + h_0$ and that of $c(x)$, where $W_0$ and $h_0$ are obtained by solving Eq. (3) with $\lambda = 0$. Concepts with low detectability, are filtered and removed.

We then train a single linear layer with the remaining concepts via the objective in Eq. (3) using the Adam optimizer [40]. The rank of $W_\lambda$ is tracked each time the linear map is updated. When the full-rank constraint is not fulfilled, we add a small perturbation to its zero singular values, as detailed in Algorithm 1. The hyper-parameter $\lambda$ is chosen adaptively to find a good trade-off between concept alignment and sparsity of concept weights, as detailed in Appendix A.3.

## 3.3. Global and local explanations

After $W_\lambda$ and $h_\lambda$ are obtained for a fixed $\lambda$, the original black-box model predictions (logits) can be reformulated–exactly–as a linear combination of $M$ interpretable concepts. Denoting $W_\lambda$ and $h_\lambda$ as $W$ and $h$ for simplicity, these are given by

$$z = \tilde{A}(Wf(x) + h) + \tilde{b} = Af(x) + b, \tag{5}$$

where $Wf(x) + h \in \mathbb{R}^M$ represents the activations in the concept space, and $\tilde{A} := AW^+$ and $\tilde{b} := b - \tilde{A}h$ provide the adapted affine classifier. Notice that, since $W^+W = I$, these logits have not changed from those in the original prediction. Yet, this expression allows us to compute the difference in importance of a given concept to a specific class. To understand the prediction of class $i$ in terms of concepts, we can compute the deviation of $z_i$ from the mean logit across classes:

$$z_i - \frac{1}{K}\sum_{j=1}^{K} z_j = \sum_{m=1}^{M} \underbrace{\left(\tilde{A}_{i,m} - \frac{1}{K}\sum_{j=1}^{K}\tilde{A}_{j,m}\right)}_{\Gamma_{i,m}} (Wf(x) + h)_m + B, \tag{6}$$

where $B = b_i - \frac{1}{K} \sum_j b_j + \frac{1}{K} \sum_j (\tilde{A}h)_j - (\tilde{A}h)_i$ is a constant term.

From this, we can identify the *global* weights of concept $s_m$ in predicting class $i$, $\Gamma_{i,m} := \tilde{A}_{i,m} - \frac{1}{K} \sum_j \tilde{A}_{j,m}$. Moreover, $\gamma_{i,m}(x) := \Gamma_{i,m}(Wf(x) + h)_m$ is the *local* contribution of concept $s_m$ to the specific sample $x$. From these, one can also calculate the difference between classes $i$ and $j$ in terms of their employed concepts:

$$z_i - z_j = \sum_{m=1}^{M} (\tilde{A}_{i,m} - \tilde{A}_{j,m})(Wf(x) + h)_m + b_i - b_j + (\tilde{A}h)_j - (\tilde{A}h)_i. \tag{7}$$

These local and global explanations will be demonstrated shortly in the Experimental Section.

### 3.4. Comparative methods

We compare our method with the following existing CBM-based approaches.

**Post-hoc CBM (PCBM) [27]** PCBM trains a linear classifier to learn concept activation vectors (CAVs) [28], and projects the image embedding onto these to obtain concept scores. In cases without concept annotations, CLIP-derived image embedding vectors are projected onto text embeddings to obtain concept scores. A sparse linear classifier is trained on these concept scores to make final predictions.

**Language in a Bottle (LaBo) [20]** LaBo is similar to the CLIP version of PCBM, but the concept score is defined as the inner product of the image and concept embeddings. A linear predictor with softmax-normalized coefficients is then trained to predict labels, with no sparsity regularization.

**Label free CBM (LF-CBM) [21]** LF-CBM first learns a linear mapping from a black-box model's latent space to a concept space and then trains a sparse linear predictor on the resulting features to predict the class labels. While the mapping from hidden space to concept space shares a similar motivation to our method, LF-CBM learns a new predictor to predict labels and, as a result, it cannot explain the original prediction and typically results in a loss of predictive power.

## 4. Experiments

**Datasets** We evaluate our method on six datasets in total: three for standard image classification sets (CIFAR-10, CIFAR-100 [41] and ImageNet [42]), and three fine-grained image classification datasets (CUB [37], AwA2 [43], and Food-101 [44]).

**Black-box models** Our method applies to any black-box model provided that the number of concepts ($M$) is larger than the dimension ($d$) of its last FCN, which is not a big restriction in practice, as we will shortly show. We train a black-box model for each dataset, with the image encoder of CLIP-ViT-L/14[26] as the backbone, and attach a two-layer multi-layer perceptron (MLP) as the classifier. The hidden dimension of MLP is set to be 64 for CIFAR-10 and AwA2, and 256 for CIFAR-100, CUB, and Food-101. For ImageNet, we train a linear probe on top of the image encoder of CLIP-ViT-L/14 as the black-box model. During training, the image encoder is fixed.

**Concept bank** Each dataset has an associated concept bank relevant to its task. We use existing concept annotations for CUB and AwA2. For CIFAR-10, CIFAR-100, and ImageNet, we curate 85, 691, and 2,901 concepts, respectively, by querying ConceptNet (a knowledge graph connecting textual concepts with edges between them) [45] with class names. For Food-100, we use 1,295 concepts curated by LaBo [20] using GPT-3, since class names are specific food names and less present in ConceptNet. We present more details about concept curation in Appendix A.3. Importantly, all methods use the same concept bank per dataset for fair comparison.

**Concept features** As described in Section 3, CLIP models are used to generate concept feature $c(x)$, which is a proxy of ground truth concept features when annotations are not available. We use the CLIP-ViT-L/14 model for the main results, and include other versions of CLIP models in Appendix

Table 1: Accuracy and Global Explanation Quality on General Image Classification sets

| Method | CIFAR-10 | | CIFAR-100 | | ImageNet | |
|---|---|---|---|---|---|---|
| | ACC | X-Fact@10 | ACC | X-Fact@10 | ACC | X-Fact@10 |
| Black- box | 0.981 | – | 0.873 | – | 0.844 | – |
| PCBM | 0.937 | $0.650 \pm 0.136$ | 0.826 | $\mathbf{0.501 \pm 0.173}$ | 0.814 | $0.307 \pm 0.158$ |
| LaBo | 0.976 | $0.540 \pm 0.080$ | 0.855 | $0.422 \pm 0.159$ | 0.830 | $0.225 \pm 0.119$ |
| LF-CBM | 0.979 | $0.670 \pm 0.142$ | 0.848 | $0.375 \pm 0.131$ | 0.708 | $0.248 \pm 0.126$ |
| CBM-zero(Ours) | $\mathbf{0.981}$ | $\mathbf{0.683 \pm 0.107}$ | $\mathbf{0.873}$ | $0.360 \pm 0.131$ | $\mathbf{0.844}$ | $\mathbf{0.334 \pm 0.163}$ |

Table 2: Accuracy and Global Explanation Quality on Fine-grained Image Classification sets

| Method | CUB | | AwA2 | | Food-101 | |
|---|---|---|---|---|---|---|
| | ACC | X-Fact@10 | ACC | X-Fact@10 | ACC | X-Fact@10 |
| Black- box | 0.861 | – | 0.981 | – | 0.953 | – |
| PCBM | 0.824 | $0.119 \pm 0.123$ | 0.632 | $0.486 \pm 0.201$ | 0.947 | $0.413 \pm 0.185$ |
| LaBo | – | – | 0.897 | $\mathbf{0.674 \pm 0.161}$ | 0.943 | $0.335 \pm 0.161$ |
| LF-CBM | 0.831 | $0.268 \pm 0.171$ | 0.976 | $0.673 \pm 0.136$ | 0.944 | $\mathbf{0.545 \pm 0.171}$ |
| CBM-zero(Ours) | $\mathbf{0.861}$ | $\mathbf{0.620 \pm 0.180}$ | $\mathbf{0.981}$ | $0.616 \pm 0.135$ | $\mathbf{0.953}$ | $0.477 \pm 0.175$ |

A.3. In the case of CUB, which has image-wise concept annotations and where general CLIP models struggle to capture these fine-grained annotated concepts (see Appendix A.4), we use the annotated presence labels as $c(x)$. For a fair comparison, we also use ground truth annotations in PCBM (the CAV version) and LF-CBM for cases where image-wise concept annotations are present. Yet, this does not apply to LaBo, as the CLIP score is essential to its input.

## 4.1. Results

**Prediction Power**  The prediction power of models is easily evaluated by their classification accuracy, as shown in Table 1 and Table 2. We obtain the highest accuracy across all the datasets as expected, since it inherently preserves the black-box models' prediction.

**Global explanations**  The evaluation of explanation quality is more challenging. Prior works have often omitted it [27], or relied on subjective human inputs [20, 21]. In this work, we define a new metric, termed *X-factuality@k*, to evaluate the validity of k-top concepts receiving the highest global weights ($\Gamma_{i,m}$) to explain a given class $i$. The definition of *validity* of concepts differs per dataset, depending on whether human annotations are present or not:

1. For concepts curated from ConceptNet, a concept is *valid* if there exists a valid edge connecting it and the corresponding class name[2].

2. For concepts collected from GPT, we prompt GPT-4 with "*Please assign a score between 0 and 1 based on the importance of {concept} in visually recognizing {class name}*". Concepts receiving scores higher than 0.5 are deemed *valid*.

3. AWA2 has class-wise concept annotations, and we consider concepts with "presence" annotations as *valid*.

4. CUB provides image-wise concept annotations, which are aggregated to class-wise continuous scores (ranging from 0 to 100), indicating the percentage of times a concept is marked as "present" within each class. We consider concepts with values above 50% as *valid*.

We denote the estimated set of top $k$ important concepts for class $i$ as $\hat{S}_i^k$, and the "valid" concepts as $S_i$. Given $\hat{S}_i^k$ and $S_i$, we define X-factuality as:

$$\text{X-factuality}_i @ k = \frac{|\hat{S}_i^k \cap S_i|}{k}, \tag{8}$$

---

[2]The ConceptNet assigns a semantic meaning to each edge, and we consider an edge valid if its semantic meaning is not "Obstructed By", "Antonym", "Distinct From", or "External URL".

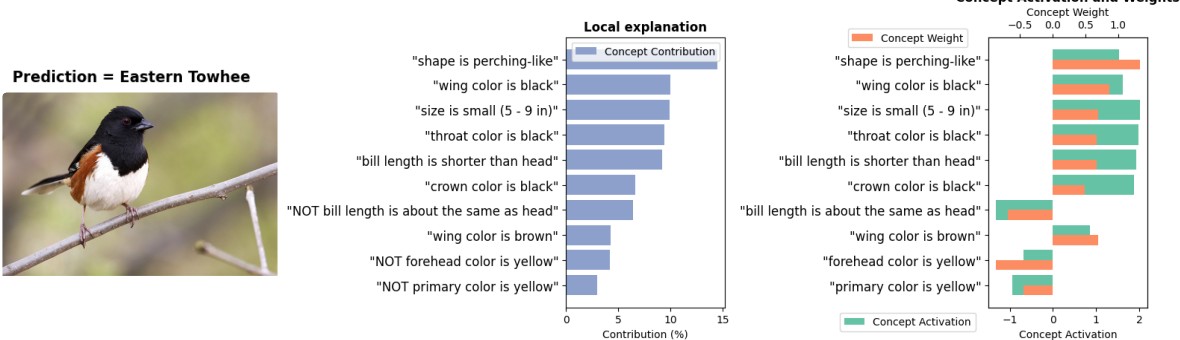

Figure 3: **An examples of local explanations.** Blue bars show the percentage concept contributions ($\gamma_{i,m}(x)$) to logit. Cyan bars show the concept activation ($(Wf(x) + h)_m$), and orange bars show the concept weights ($\Gamma_{i,m}$). Negative values of both activation and weight indicate the absence of certain concepts contributing to prediction, and we prepend "NOT" to the concept name to reflect this.

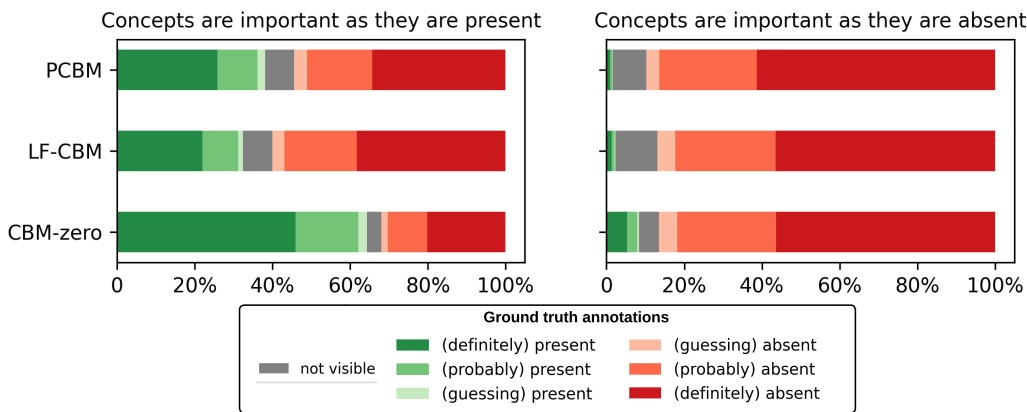

Figure 4: **Quantitative assessment of local explanations quality on CUB.** The top 10 contributing concepts per image are selected and compared with ground truth, which has 7 possible certainty-presence combinations: definitely/probably/guessing for either present or absent, or not visible). These top concepts can be important due to their presence or absence. We calculate the proportion of these 7 combinations for both, concepts that are present (left, more green is better) and absent (right, more red is better).

Note that while the definition of X-factuality seems related to the commonly used precision, they are not equivalent since "valid" concepts are not necessarily important for a prediction task. For instance, "fur" might be "valid" in many animals, but might not be important in classifying different types of animals. A high X-factuality only indicates the selected concepts do not contradict human intuition, but does not guarantee the correctness of an explanation, since the actual concept importance for predictions is not always clear or well-defined.

Table 1 and Table 2 show results on X-Factuality@10 of global explanation for general and fine-grained tasks, respectively. X-factuality is calculated per class and aggregated as mean and standard deviation. CBM-zero obtains comparable interpretation quality with other methods, ranking first or second in most cases. Figure A.1 further shows X-factuality@$k$ as a function of $k$.

**Local explanations** Recall from Sec. 3.3 that the local contributions of concept $s_m$ for predicting the sample $x$ as class $i$ is defined as $\gamma_{i,m}$. Figure 3 illustrates the top 10 contributing concepts and their percentage of contribution to class logit for an example image from CUB. Note that concepts with high contribution are not always *present* in certain images. Conversely, the absence of a concept

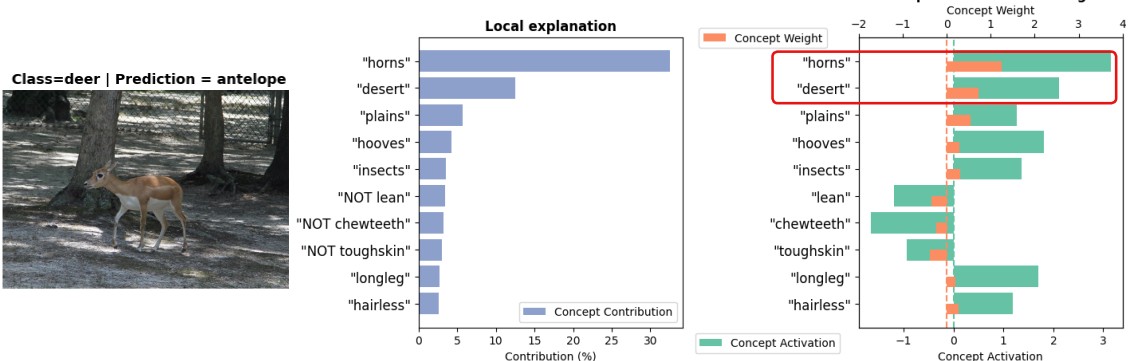

Modify concept activation:  "horns": 3.15 -> 0, "desert": 2.11 -> 0                    Prediction: "antelope" -> "deer"

Figure 5: **An example of manual intervention.** The image gets an incorrect prediction of "antelope". By zeroing out the unreasonably high activations of "horns" and "desert", the prediction is corrected.

can also contribute towards a predicted label, when both $\Gamma_{i,m}$ and $(Wf(x)+h)_m$ are negative. We prepend "NOT" to the concept in this case. More examples appear in Appendix A.4.

Quantitative evaluation of these local explanations needs costly sample-wise concept annotations, which are only available for CUB. We select the top 10 contributing concepts per image (estimated by CBMs) and compare them with annotations, which have 7 possible presence-certainty combinations. As mentioned before, a concept's contribution can be caused by its presence or absence, depending on whether its activation is positive or negative. We assess alignment with ground truth annotations for both cases and present these results in Figure 4. All the methods perform well in identifying concepts contributing by their absence, with more than 80% of them annotated as absent. Our method performs best in identifying concepts contributing by their presence, with 42.6% annotated as "definitely present" and another 16.6% annotated as "probably present", significantly better than comparative methods.

**Interventions**   Like other CBM-based methods, CBM-zero supports human intervention. Incorrect predictions can be fixed by identifying and manually editing errors in concept activations $(Wf(x) + h)$ using domain knowledge, known as local interventions. Notably, intervening in CBM-zero is equivalent to intervening in the original black-box model, as these two are strictly equivalent in their formulations. We show an example of manual intervention in Figure 5, where an image of "deer" is incorrectly predicted as an "antelope". By inspecting the local explanations, we identified that the top contributed concepts – "horn" and "desert"– get high activations, which are inconsistent with the image. By zeroing out the activation of these two concepts, the prediction is successfully corrected. We show more examples of such local interventions in Figure A.13 and A.14. In Appendix A.5, we also include global intervention studies, where we modify concept weights $(\Gamma_{i,m})$ to change model behavior and gain improvements globally.

## 5.  Discussion, Limitations, and Conclusion

In this work, we introduce CBM-zero, which explains black-box models by constructing a CBM via an invertible mapping between its latent space and an interpretable concept space. Unlike other CBMs, CBM-zero does not alter the original black-box model, preserving its performance exactly. Experiments across various benchmarks demonstrate its superiority in maintaining the model's accuracy and providing high-quality interpretations compared with states-of-arts. This work also has several limitations. The affine mapping might not always be powerful enough, and leveraging more expressive yet still invertible models (e.g., normalizing flows [46]) in the future might address this. The reliance on CLIP models introduces limitations. CLIP scores measure correlation rather than directly indicating the presence of concepts. Therefore, an image of a baby might get a high score for "stroller" even if no stroller is shown in the image. Exploring alternatives, such as querying large language and vision models [38] or human collaborators [47] about the presence of certain concepts, might address this issue. Finally, the concept banks used are not perfect. ConceptNet,

although large and evolving, is not sufficient to represent the numerous concepts and complex relationships in the real world. GPT-generated concepts are more flexible but might be overly complex and include non-visual descriptions. Both ConceptNet and GPT-generated concepts can be inaccurate in describing the relationship between concepts, with unclear levels of noise. Human annotations are costly and labor-intensive, hindering its broad applicability. Incorporating recent advances in concept discovery [22–24, 48], as well as incorporating notions of uncertainty quantification to our results [49, 50], may help generate more reasonable explanations.

## 6. Reproducibility Statement

Codes for implementing this method and reproducing the results can be found in this repository: `https://github.com/JasmineZhen218/CBM-zero`

## 7. Acknowledgment

This research was supported by NSF CAREER Award CCF 2239787.

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

# A. Appendix

The Appendix provides additional clarifications and details that are not included in the main text due to the length limit.

## A.1. X-factuality@$k$ as a function of $k$

In the main text, we only report X-factuality @ 10 for global explanations in Table 1 and 2. Here, we extend this by plotting X-factuality@$k$ as a function of $k$, as shown in Figure A.1.

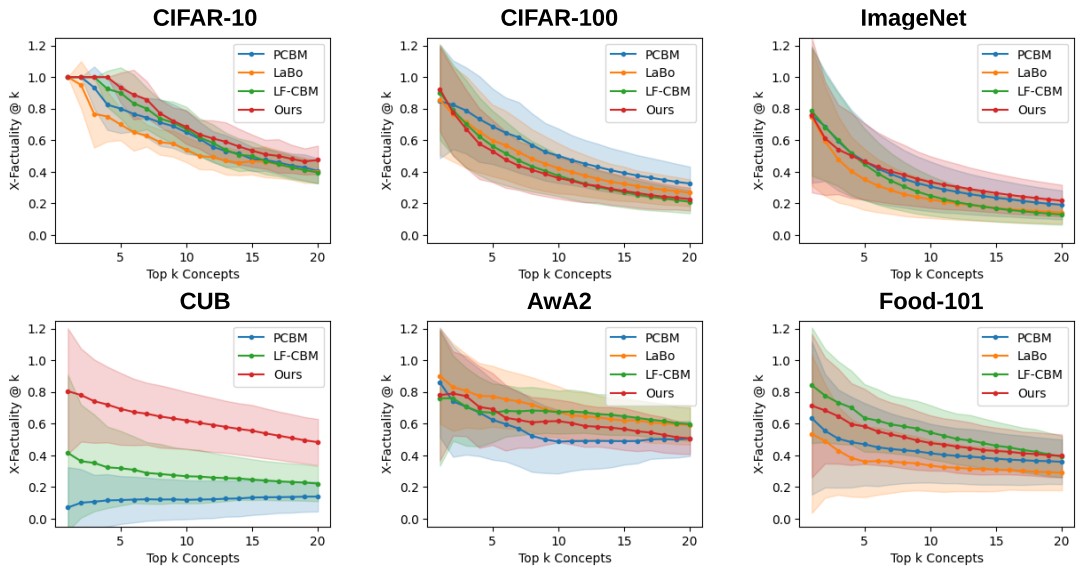

Figure A.1: X-Factuality @ $k$ v.s. $k$. Solid lines show the mean X-Factuality across classes, and the shaded area shows the standard deviation among classes.

## A.2. CLIP models on CUB

As described in Section 4, the CUB dataset contains 312 human-annotated concepts. These concepts are fine-grained descriptions of the color, shape, and size of specific bird parts. We found that even state-of-the-art CLIP models struggle to align these concepts with the correct images, reducing the faithfulness of any explanation methods relying on them. To demonstrate this, we select the top 10 concepts with the highest CLIP scores per image and compare them with annotations. Table A.1 summarizes the results of different versions of CLIP models, and none of them align with the ground truth well.

## A.3. Implementation details

**Concept curation** For CIFAR-10, CIFAR-100, and ImageNet, we collect concepts by querying ConceptNet [45] with the {*class name*} and obtain {*concepts*}). Only concepts with valid connections (excluding connections with semantic meaning of "Obstructed By", "Antony", "Distinct From", or "External URL") are retained. For a specific dataset, the concepts connecting all the class names are gathered, and the following processing is applied: (1) long concepts with more than 10 characters are excluded to include simple concepts instead of complex statements; (3) only the top 10 concepts with the highest connection strength to each class name are preserved, which excludes those less common, and weakly related concepts; (3) concepts that are close to each other and close

Table A.1: Quality of different versions of CLIP models on capturing concepts of CUB

| Annotations | | Composition of Top-10 Correlated Concepts (%) | | | | |
|---|---|---|---|---|---|---|
| Presence | Certainty | RN50 | RN101 | ViT-B/16 | ViT-B/32 | ViT-L/14 |
| | Definitely | $7.71 \pm 11.5$ | $6.69 \pm 10.5$ | $8.97 \pm 13.0$ | $8.45 \pm 11.8$ | $10.4 \pm 12.3$ |
| Yes($\uparrow$) | Probably | $4.93 \pm 9.04$ | $3.73 \pm 7.92$ | $5.04 \pm 9.74$ | $5.43 \pm 9.76$ | $5.86 \pm 9.72$ |
| | Guessing | $1.43 \pm 4.43$ | $0.97 \pm 3.73$ | $1.17 \pm 4.41$ | $1.54 \pm 4.93$ | $1.44 \pm 4.60$ |
| | Definitely | $39.3 \pm 31.6$ | $45.6 \pm 33.7$ | $41.0 \pm 32.1$ | $35.7 \pm 32.8$ | $38.1 \pm 31.6$ |
| No($\downarrow$) | Probably | $23.0 \pm 27.1$ | $25.4 \pm 29.0$ | $23.1 \pm 27.3$ | $21.7 \pm 27.8$ | $21.8 \pm 27.5$ |
| | Guessing | $6.37 \pm 15.3$ | $6.10 \pm 15.4$ | $5.66 \pm 14.6$ | $6.25 \pm 16.1$ | $6.05 \pm 15.9$ |
| Not visible | | $17.3 \pm 24.0$ | $11.6 \pm 18.7$ | $15.1 \pm 22.5$ | $20.9 \pm 29.6$ | $16.4 \pm 25.8$ |

We select the top 10 concepts with the highest CLIP scores per image, calculate their proportions of 7 possible presence-certainty combinations and report the mean and standard deviation across images.

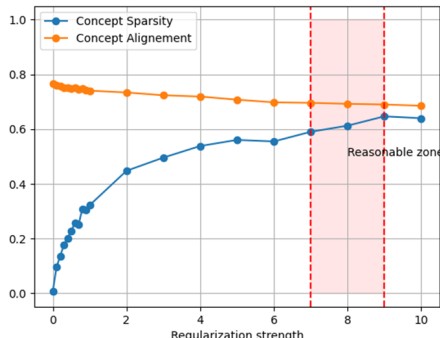

Figure A.2: An illustration of the selection of regularization strength $\lambda$. The x-axis shows values of $\lambda d$, where $d$ is the dimension of $f(x)$.

to class names are excluded, where the text-text similarity is measured by cosine similarities of the embedding encoded by *all-mpnet-base-v2* [51] sentence encoder and a threshold of 0.85 is applied. For Food-101, we use the concepts curated by LaBo [20], where the GPT-3 is prompted with *describe what {class name} looks like* and relevant concepts are extracted from the answers. We exclude overly long and complex concepts with more than 15 characters.

**Hyperparameters** $\lambda$  The regularization strength, $\lambda$, controls the trade-off between concept alignment and the sparsity of concept weights, which are both important to generate good explanations. Given a specific $\lambda$, we quantitatively assess them in the following ways:

1. concept alignment: We calculate the Pearson correlation coefficient between the $i$th entry of $W_\lambda f(x) + h_\lambda$ and that of $c(x)$, resulting in the alignment of the $i$th concept. We calculate the average of alignments across concepts.

2. concept sparsity: we assess the ratio of zero values in $AW_\lambda^+$ with the cut-off of 0.01.

We search $\lambda$ in a range of $[0, \frac{10}{d}]$ ($d$ is the dimension of $f(x)$) and monitor the change of concept alignment and concept sparsity. Naturally, the alignment of the concept decreases and the sparsity increases as $\lambda$ increases. We aim to encourage sparsity without compromising concept alignment too much. Practically, we decide on a "reasonable zone" for $\lambda$, where the mean alignment of the concept is not less than 90% of its maximum, and the sparsity is not less than 90% of its maximum during the search. If the reasonable zone covers more than one choice of $\lambda$, we choose the smallest $\lambda$. Figure A.2 shows an illustration of this process, using experiments on AwA2 as an example. In this case, $\lambda = \frac{7}{d}$ is the final selection. We evaluated the alignment and sparsity of the concept of the chosen $\lambda$ and compared them with the results of not applying regularization. Figure A.3 shows the distribution of the magnitudes of concept weights and concept alignments. The results show that we can significantly increase sparsity without harming alignment by choosing a reasonable $\lambda$.

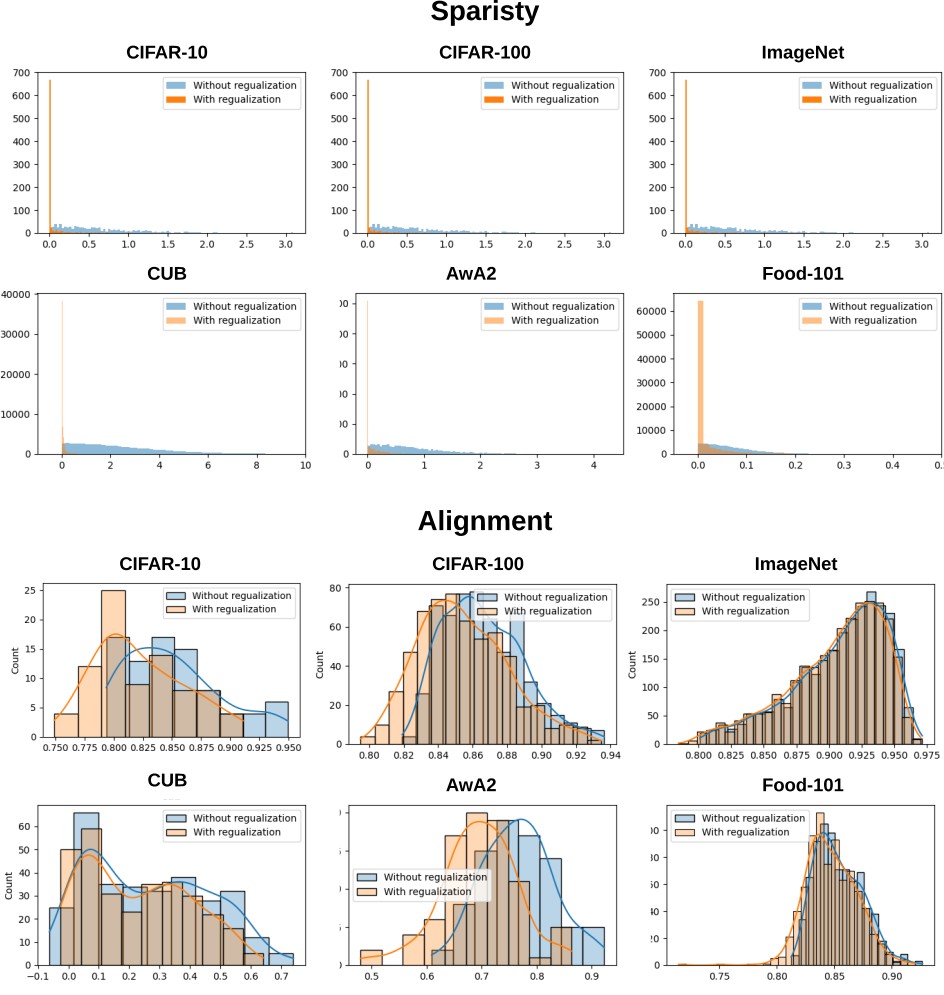

Figure A.3: Comparison of concept sparsity and alignment with and without regularization. The upper part shows the histogram of absolute values of concept weights. The lower parts show the histogram of alignments across concepts.

Table A.2: X-Factuality@10 with different choices of exponential power $t$

|  | CIFAR-10 | CIFAR-100 | ImageNet | AwA2 | Food101 |
|---|---|---|---|---|---|
| t=1 | $0.700 \pm 0.107$ | $0.355 \pm 0.131$ | $0.284 \pm 0.156$ | $0.576 \pm 0.177$ | $0.435 \pm 0.177$ |
| t=3 | $\mathbf{0.767 \pm 0.125}$ | $0.357 \pm 0.120$ | $\mathbf{0.337 \pm 0.169}$ | $0.582 \pm 0.161$ | $0.468 \pm 0.190$ |
| t=5(default) | $0.683 \pm 0.107$ | $\mathbf{0.360 \pm 0.131}$ | $0.334 \pm 0.163$ | $\mathbf{0.616 \pm 0.135}$ | $\mathbf{0.477 \pm 0.175}$ |

**Hyperparameters** $t$ As described in Section 3.2, we apply an exponential transformation with power $t$ on the CLIP scores to emphasize concepts with higher correlation to images. We set $t = 5$ as the default and discuss how the choice of $t$ affects the quality of the interpretation. Table A.2 reports the X-factuality@10 for each dataset. The results are not very sensitive to the choice of $t$, while applying exponential power ($t = 3$ or $5$) is generally better than not (i.e., $t = 1$). Note that the choice of $t$ will not affect the accuracy of classification models, as CBM-zero does not alter the original classifier by design.

Table A.3: X-Factuality@10 with different choices of CLIP models

| | CIFAR-10 | CIFAR-100 | ImageNet | AwA2 | Food101 |
|---|---|---|---|---|---|
| ViT-B/16 | **0.833 $\pm$ 0.094** | 0.315 $\pm$ 0.142 | 0.222 $\pm$ 0.126 | 0.594 $\pm$ 0.130 | 0.441 $\pm$ 0.190 |
| ViT-B/32 | **0.833 $\pm$ 0.137** | 0.310 $\pm$ 0.118 | 0.202 $\pm$ 0.124 | 0.596 $\pm$ 0.155 | 0.471 $\pm$ 0.179 |
| ViT-L/14 | 0.683 $\pm$ 0.107 | **0.360$\pm$ 0.131** | **0.334 $\pm$ 0.163** | **0.616 $\pm$ 0.135** | **0.477 $\pm$ 0.175** |

**Other CLIP versions** In main texts, we use ViT-L/14 as the clip model to estimate the presence of concepts in images. Here, we analyze the quality of global interpretations of other versions of CLIP in Table A.3.

**Computational efficiency** All the models are trained on a single Nvidia GPU. The training of a black-box model takes from a few minutes to two days depending on the dataset size. Once the feature embedding of the black-box model and CLIP image encoders are saved, the training of the affine mapping for interpretation purposes is typically within 10 minutes.

## A.4. More examples of explanations

In this section, we provide some examples of contributions of concepts in the prediction of specific images across different datasets. To be more specific, we focus on the deviation of logit of the predicted class from the mean logit across classes, $z_{\hat{y}} - \frac{1}{K}\sum_{i=1}^{K} z_i$, and calculate the contribution of concept $s_m$, denoted as $((AW^+)_{\hat{y}m} - \frac{1}{K}\sum_i (AW^+)_{im})(f(x) + h)_m$ in percentage. The top 10 contributed concepts for each image are shown. Figure A.4 and Figure A.5 show examples from CIFAR-10 and CIFAR-100. Figure A.6 shows examples from CUB, Figure A.7 shows examples from AwA2, and Figure A.7. For ImageNet with a lot of images from diverse classes, we show examples by category, furniture in Fig A.9, animals in Fig A.10, clothes in Fig A.11, and locations in Fig A.12.

## A.5. Interventions

In this section, we describe the intervention studies in more detail. The users can intervene in concept bottleneck models by identifying and manually editing errors in concept activations $(Wf(x) + h)$ and concept weights $(\Gamma_{i,m})$. We refer to the former editing as "local intervention", as it changes the prediction of individual samples at test time; and the latter editing as "global intervention", as it changes model parameters and thus impacts model behavior globally.

**Local intervention** we inspect local explanations of images where the model made wrong predictions, manually edit incorrect/counter-intuitive activations of concepts $(Wf(x)+h)$, and eventually change the logits and predictions. Figure A.13 and A.14 shows representative examples across different datasets.

**Global intervention** Like other related works [21, 27], we manually identify concepts that are important for a particular class while having minimal impact on others, and then manually edit their weights accordingly. Through a quick exploration in ImageNet, especially in classes where the original model does not perform very well, we have found several beneficial edits, as listed below.

1. If we increase the weight of the concept "seaport" for the class "deck" by 0.2, the class-wise accuracy of "deck" increases from 64% to 88%. (This also doesn't negatively affect the model performance in other classes either; in fact, the overall accuracy improves from 84.39% to 84.40%. Note that one class only constitutes a small portion of the entire dataset.)

2. If we increase the weight of the concept "reservoir" for the class "dam" by 0.2, the class-wise accuracy of "dam" increases from 88% to 92%. The overall accuracy remains.

3. If we increase the weight of the concept "screen" for the class "firescreen" by 0.2, the class-wise accuracy of "firescreen" increases from 68% to 74%. The overall accuracy improves from 84.39% to 84.41%.

4. If we increase the weight of the concept "soup" for the class "hot pot" by 0.2, the class-wise accuracy of "hot pot" increases from 78% to 84%. The overall accuracy remains.

5. If we increase the weight of the concept "kitchen" for the class "dishwasher" by 0.2, the class-wise accuracy of "dishwasher" increases from 68% to 76%. The overall accuracy improves from 84.39% to 84.40%.

Aside from labor-intensive and subjective manual global intervention, we also have utilized our definition of "valid" concepts to intervene in the model. Specifically, we iterated through all the classes and checked their most (positively) important concepts. If a concept is designated as a "valid" concept for a certain class, we increase its weight for that class by 0.05. In this manner, even without careful manual tuning, the accuracy of ImageNet has been improved from 84.39% to 84.42%.

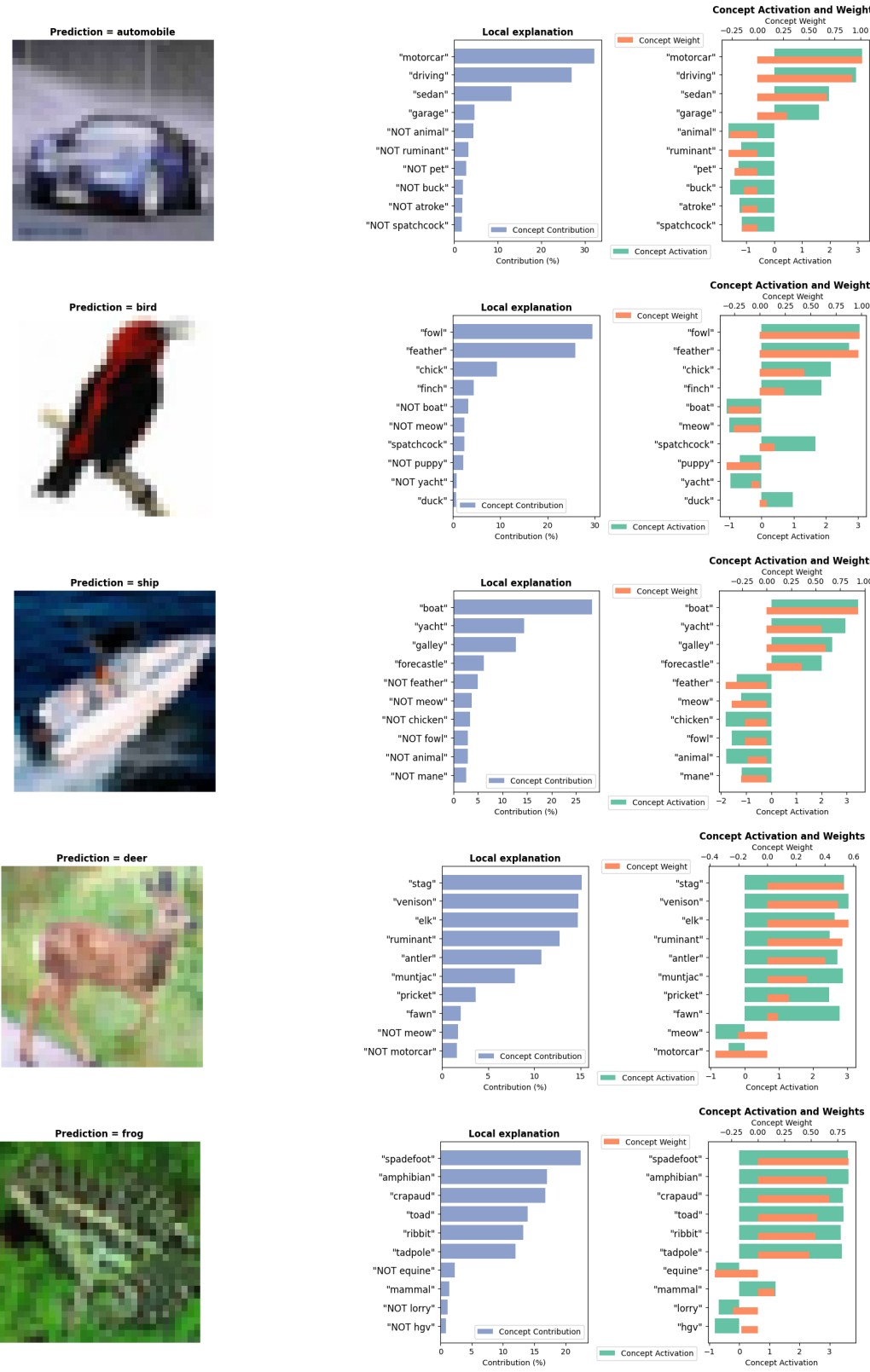

Figure A.4: Examples of local explanations from CIFAR-10. The top 10 contributed concepts are shown.

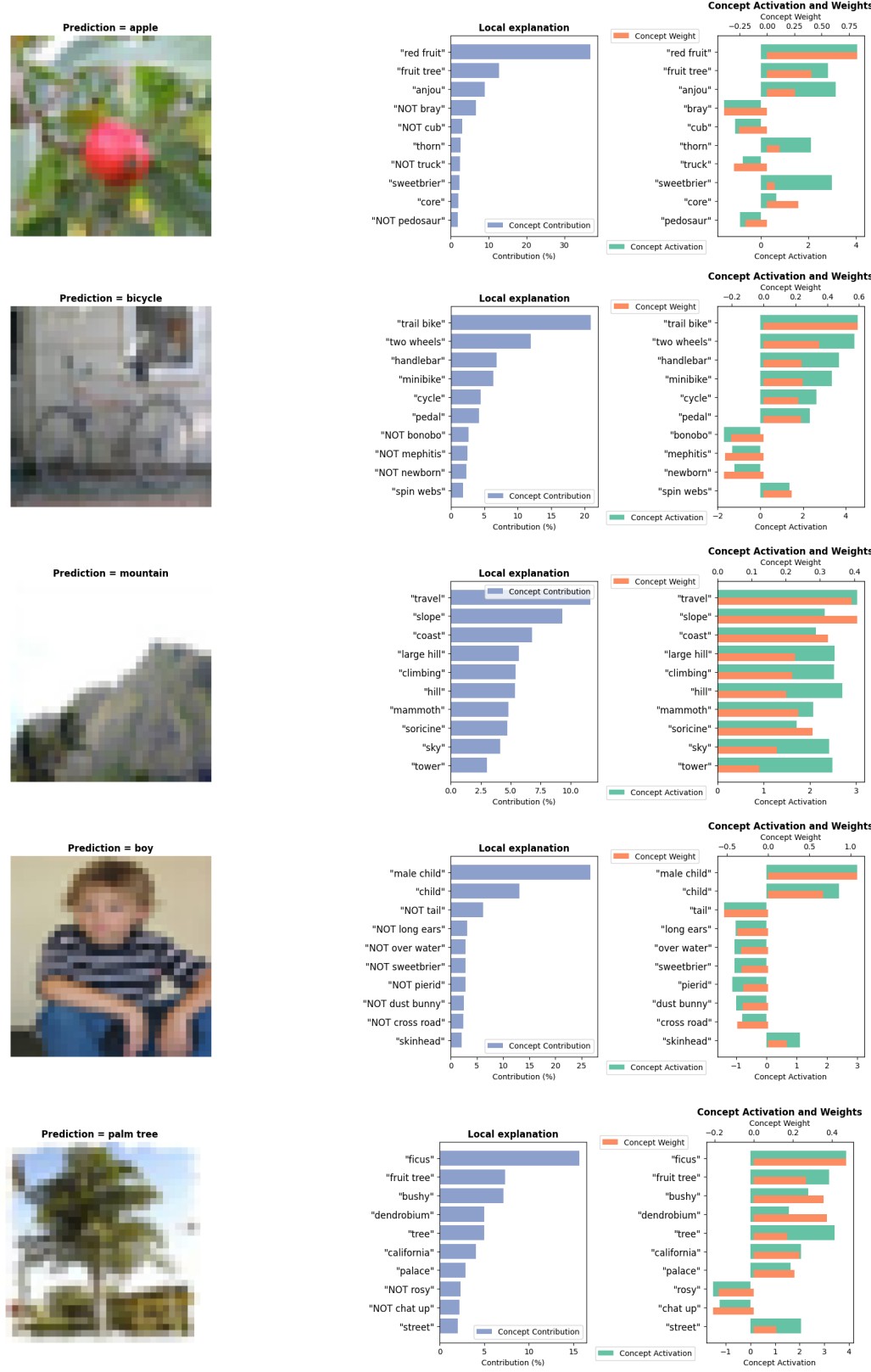

Figure A.5: Examples of local explanations from CIFAR-100. The top 10 contributed concepts are shown.

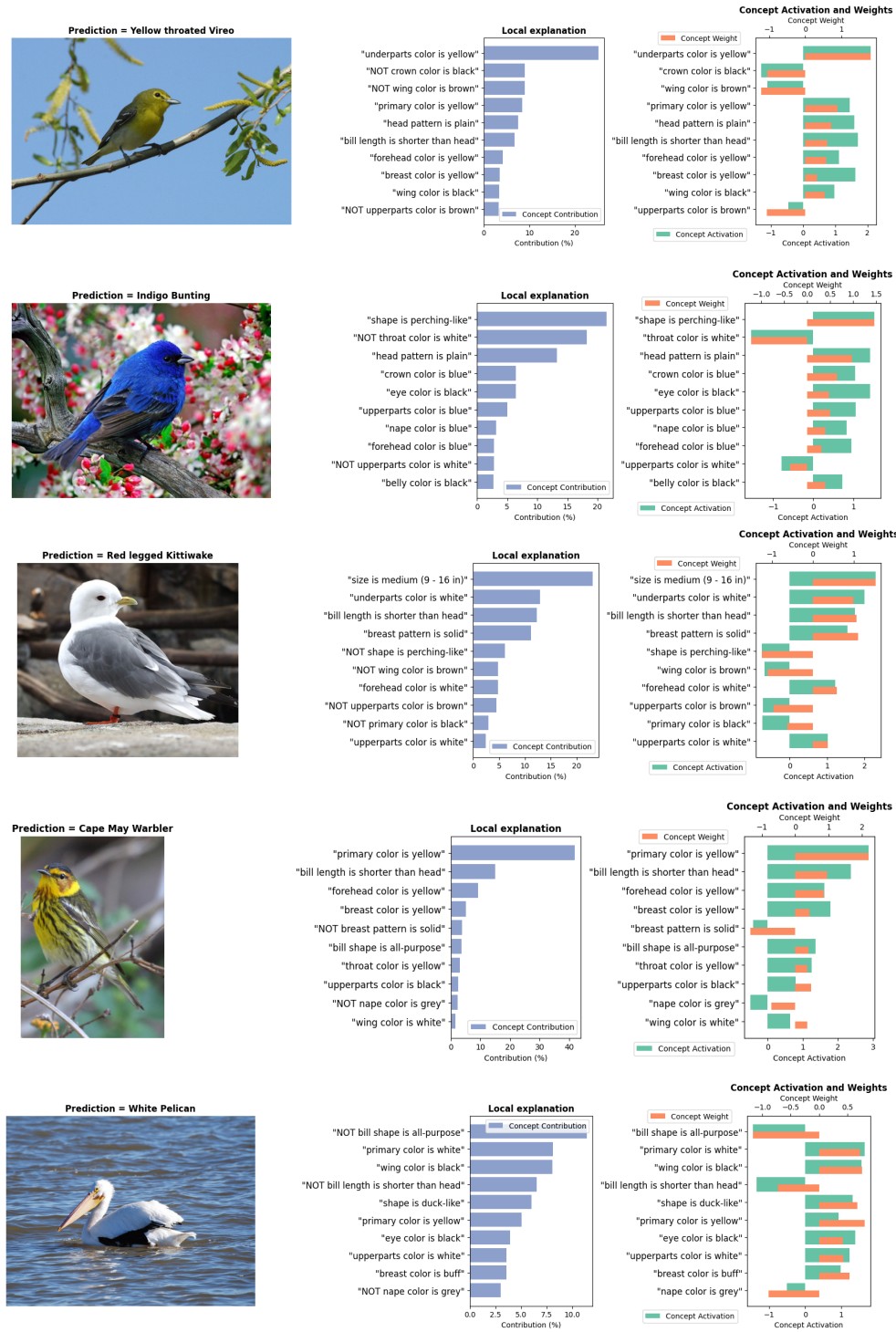

Figure A.6: Examples of local explanations from CUB. The top 10 contributed concepts are shown.

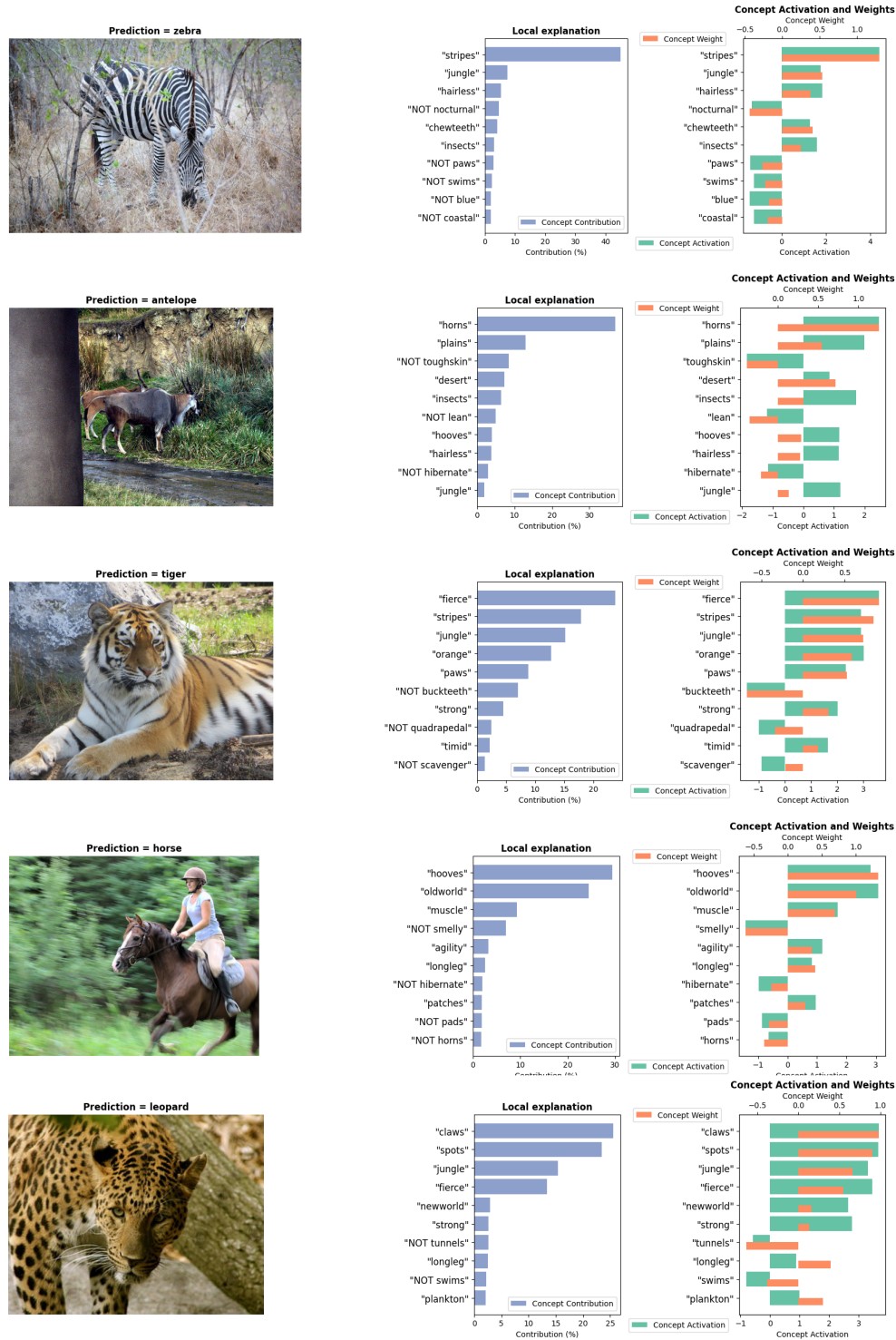

Figure A.7: Examples of local explanations from AwA2. The top 10 contributed concepts are shown.

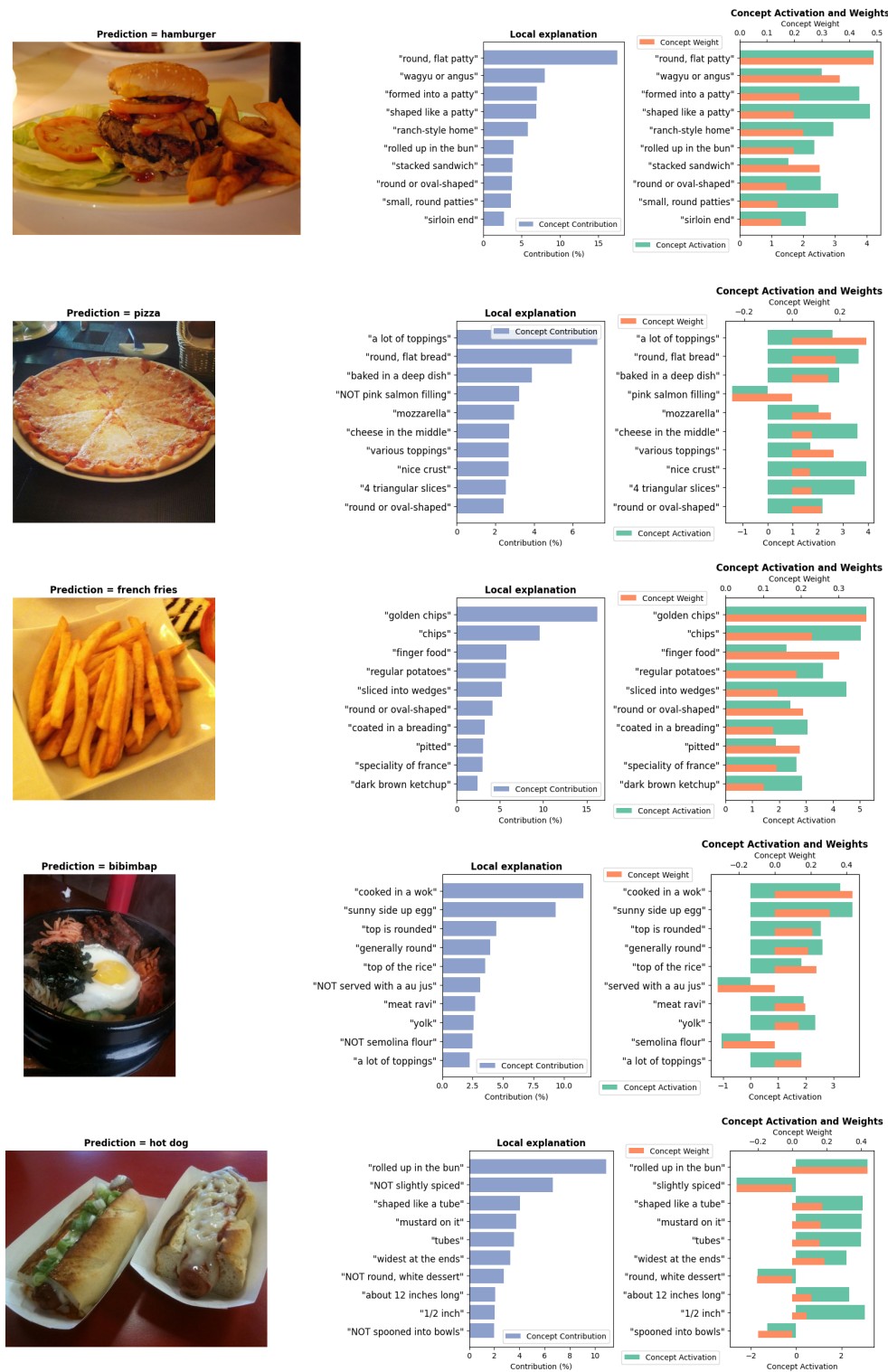

Figure A.8: Examples of local explanations from Food101. The top 10 contributed concepts are shown.

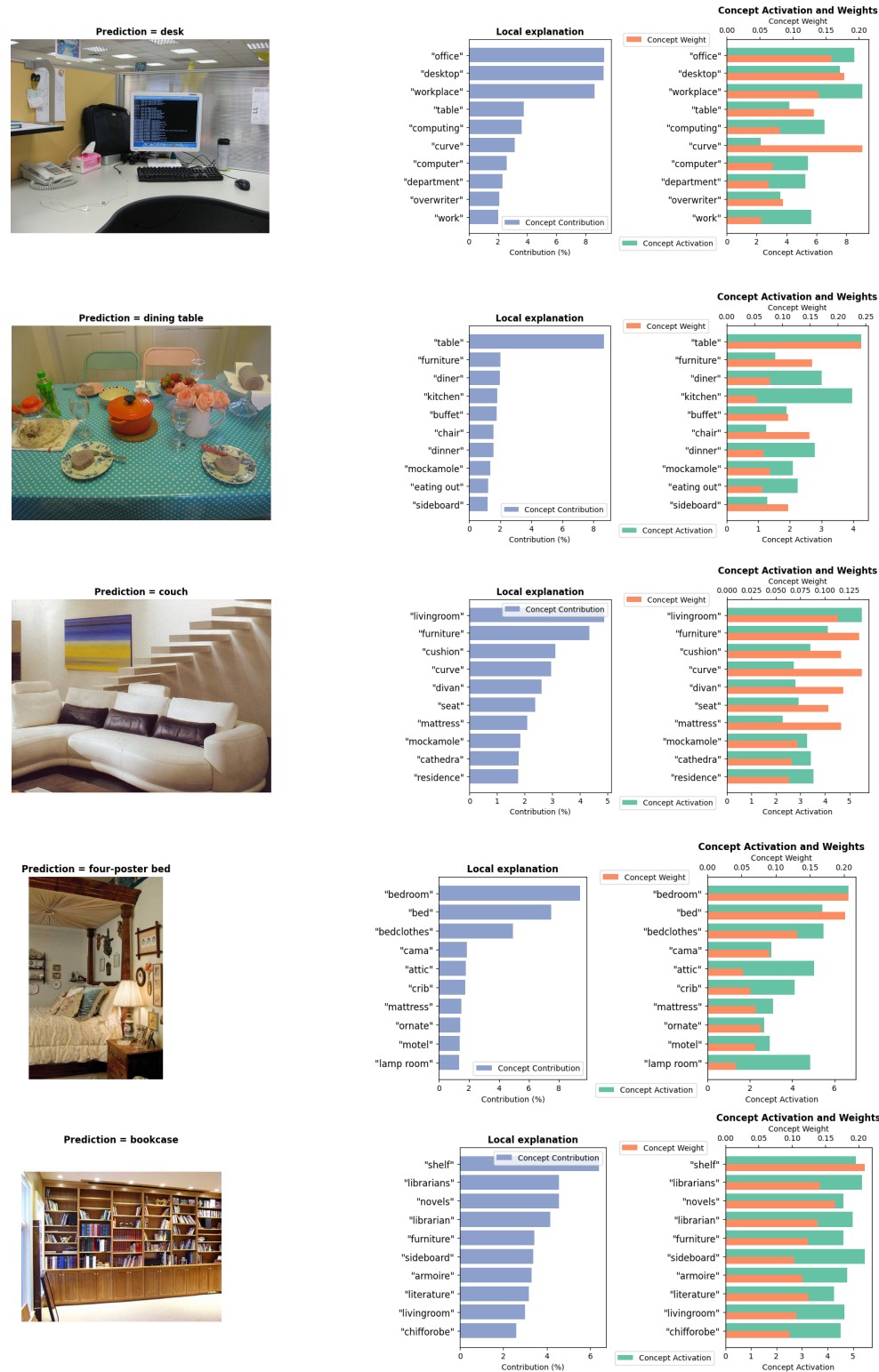

Figure A.9: Examples of local explanations from ImageNet (Furniture). The top 10 contributed concepts are shown.

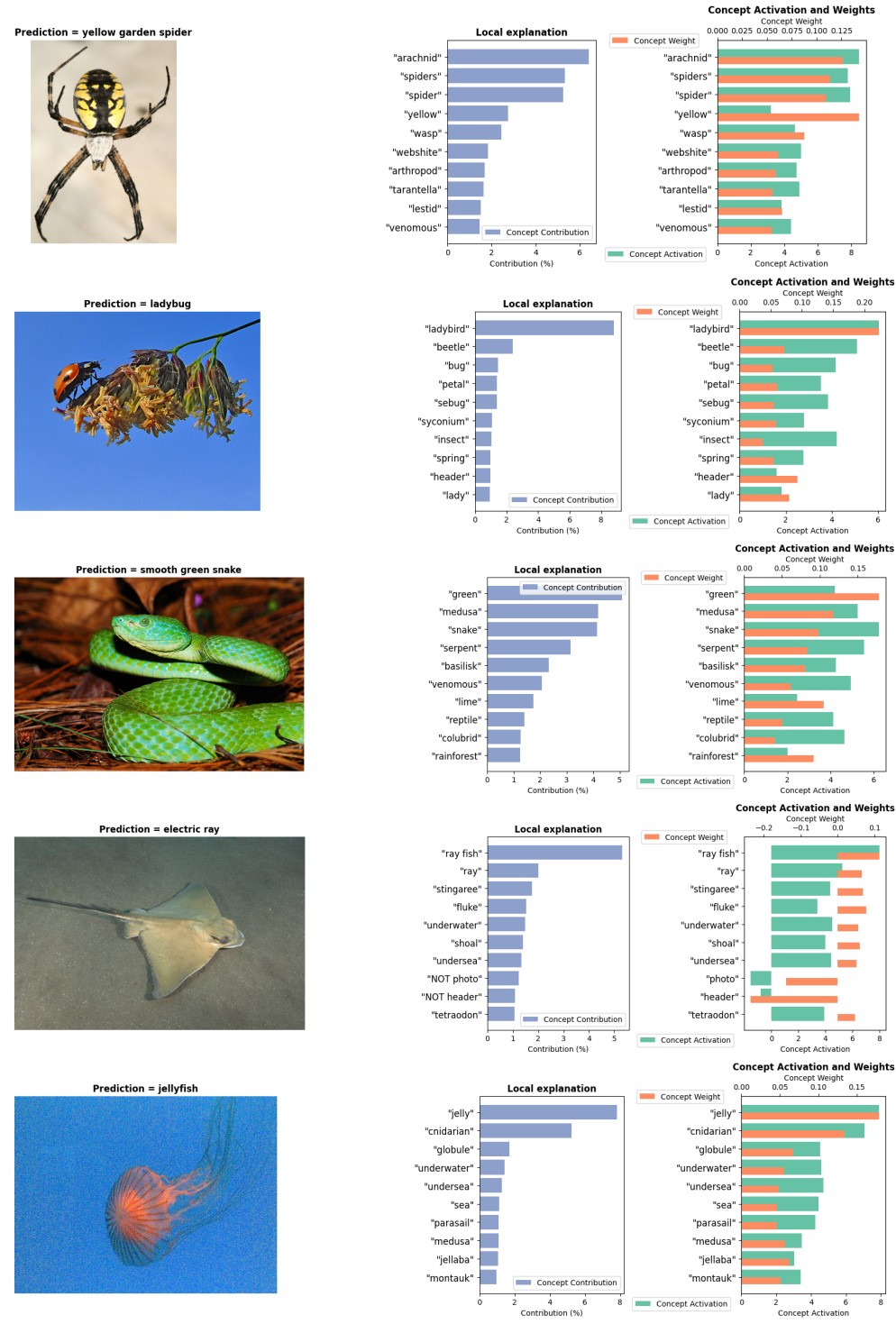

Figure A.10: Examples of local explanations from ImageNet (Animals). The top 10 contributed concepts are shown.

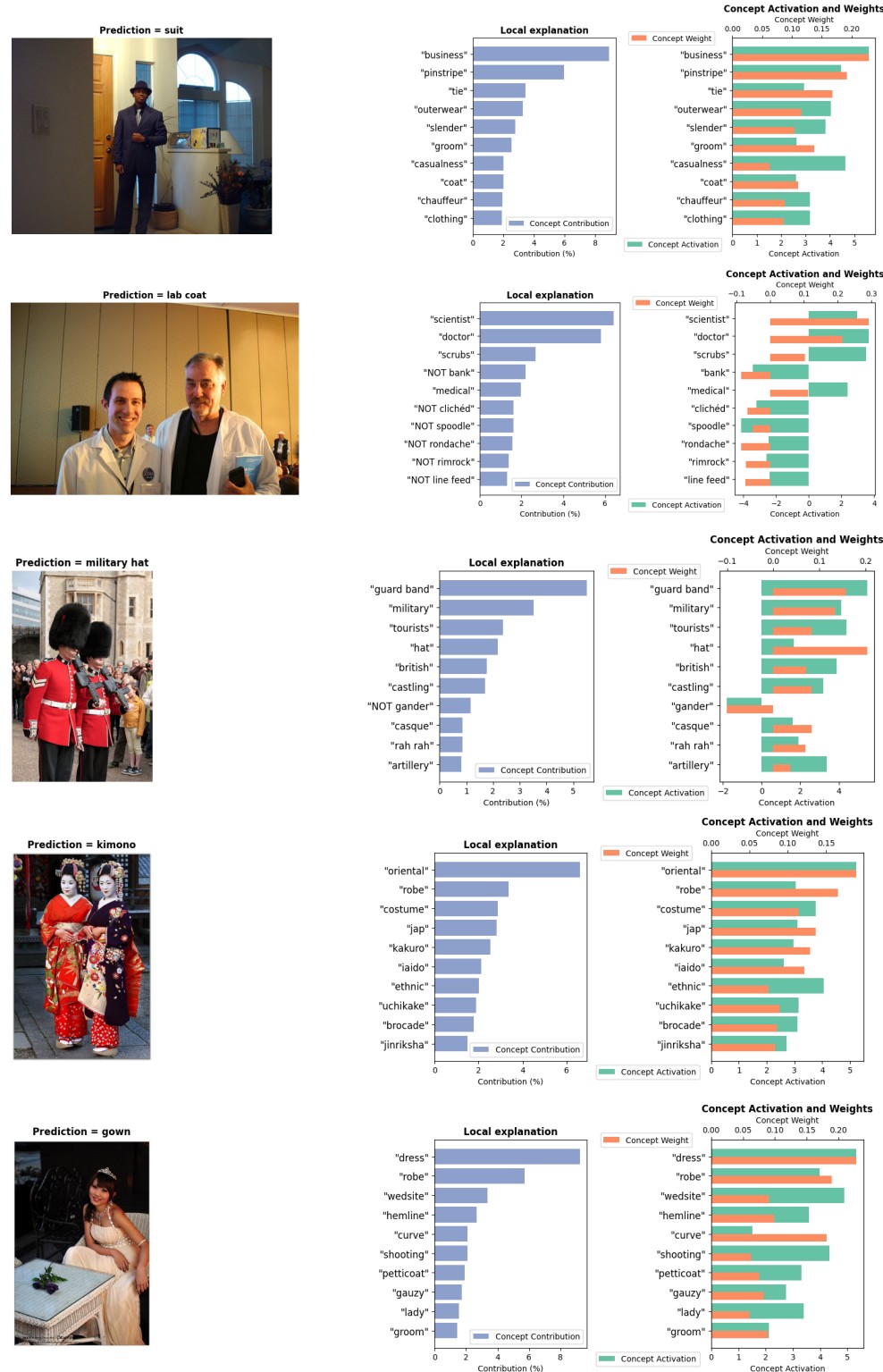

Figure A.11: Examples of local explanations from ImageNet (Clothes). The top 10 contributed concepts are shown.

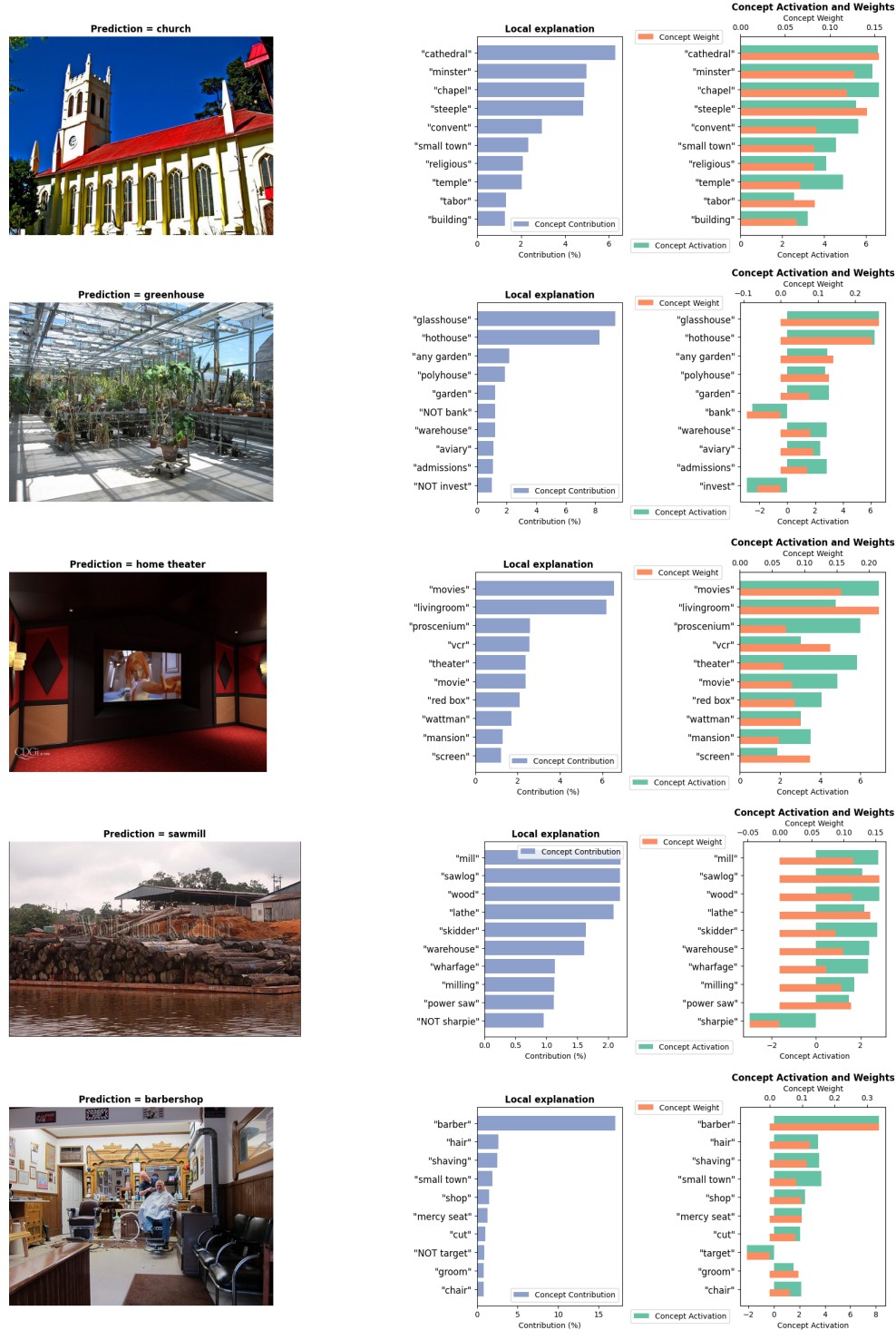

Figure A.12: Examples of local explanations from ImageNet (Locations). The top 10 contributed concepts are shown.

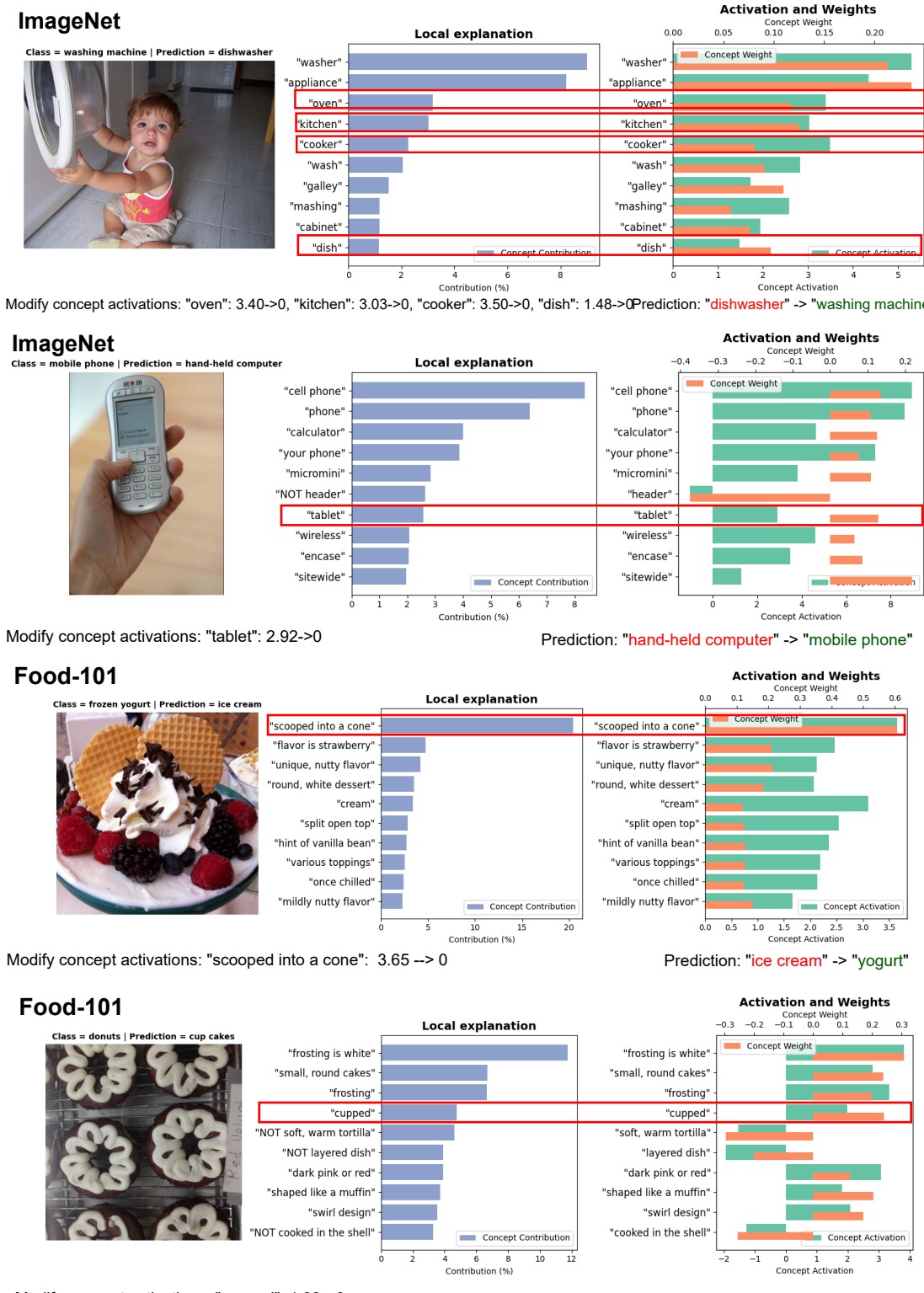

Figure A.13: Examples of local interventions.

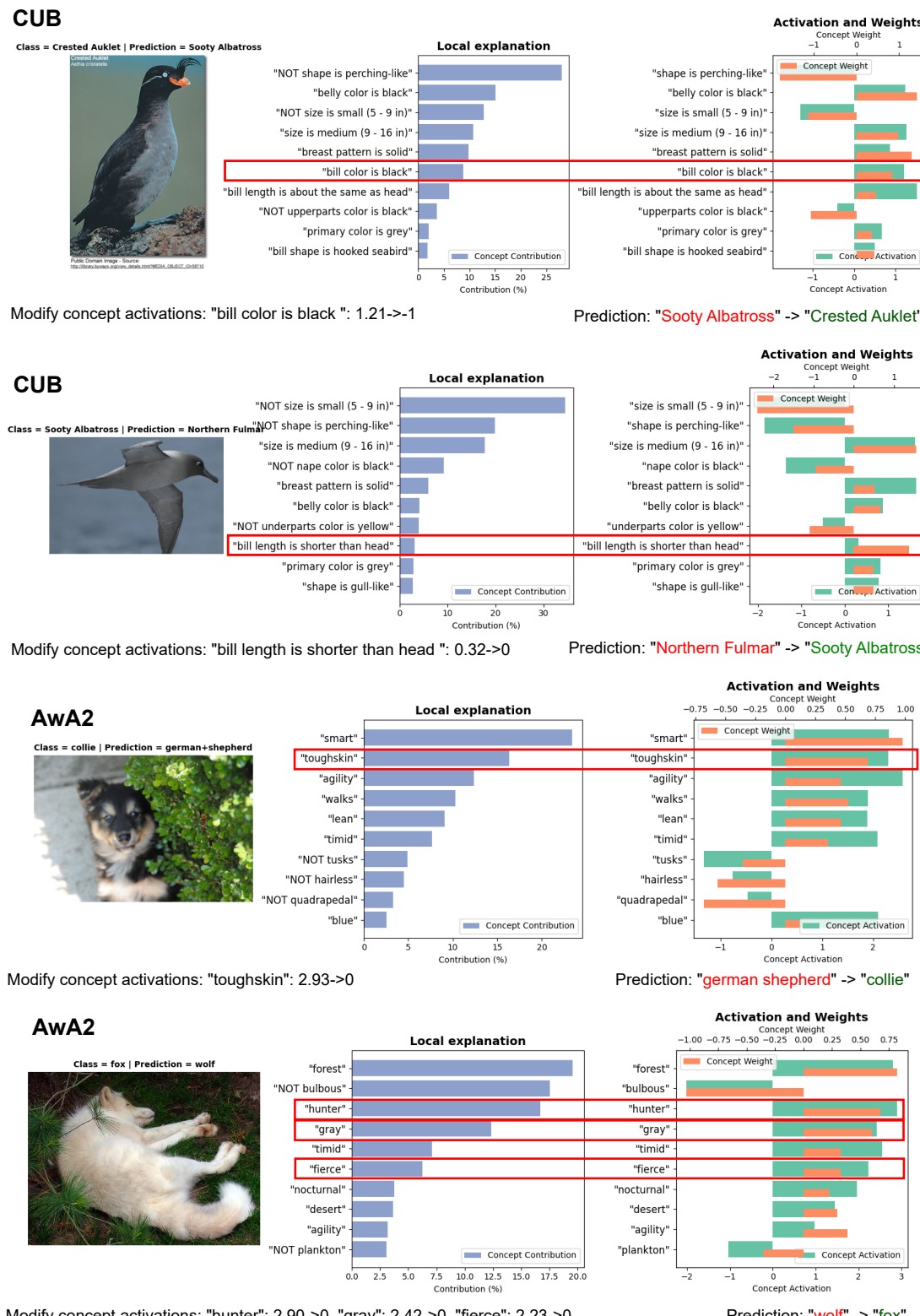

Figure A.14: Examples of local interventions.

