# OpenReview forum: "Concept Bottleneck Model with Zero Performance Loss"
_CPAL.cc/2025/Proceedings_Track — CPAL 2025 (Proceedings Track) Poster_

### Official Review · Reviewer_P2uh · 2024-12-30

**Rating:** 6
**Confidence:** 3

**Review:**

### Summary

This paper points out the weaknesses of the Concept Bottleneck Model (CBM), which often loses some prediction accuracy compared to standard black-box models. It introduces a method to convert a black-box model into a CBM without affecting its predictions or reducing its accuracy. Experiments on various datasets show that CBM-zero offers similar explainability and higher accuracy than other CBM approaches.

### 1. Quality
The paper presents an approach that effectively converts black-box models into CBMs without sacrificing accuracy.


### 2. Clarity
The paper is easy to follow, with a straightforward proposed method.

### 3. Originality
This paper ensures the invertibility of the mapping, enabling interpretations without compromising performance. It also introduces a new metric to evaluate the explanation quality.


### 4. Significance
This paper presents a method to convert a black-box model into a CBM without affecting its predictions or compromising accuracy. However, the justification for converting to CBM is unclear, especially since it does not enhance accuracy compared to black-box models. The paper does not provide sufficient explanation on this point.

### Conclusion

- **Strengths**
  - The method is solid, and experiments show promising results.

- **Weaknesses**
  - The new metric (X-factuality@k) needs validation to ensure it’s not biased toward the proposed method.
  - The method is only tested on one architecture (ViT-L-14, CLIP), so it’s unclear if it applies to other models.

---

### Official Review · Reviewer_RKJw · 2025-01-10
**Lack of Comprehensive Intervention Experiments**

**Rating:** 6
**Confidence:** 3

**Review:**

This paper presents a method to convert black-box models into concept bottleneck models (CBMs) without performance loss. It trains an invertible mapping from the latent space to the concept space while avoiding degradation of the original model's performance.

**Strengths:**

1. The paper is well-written; the concept is clearly explained and easy to understand.

2. One of the main limitations of conventional CBMs is their loss of accuracy. However, the numerical results presented in this paper show that CBM-zero achieves performance comparable to that of the original black-box model.

**Weaknesses:**

1. My main concern is the lack of comprehensive intervention testing. The authors provide only a single sample in Figure 5, which is insufficient to fully distinguish a CBM from a multi-task model.

---

### Meta-Review · Area_Chair_Vaiz · 2025-02-02

**Recommendation:** Accept (Poster)
**Confidence:** 2

**Metareview:**

Both reviewers seem to agree that the paper offers a novel method to convert black-box models into concept bottleneck models without sacrificing accuracy—a key advancement over existing CBMs. While concerns about the thoroughness of intervention experiments and the validity of the new metric have been raised, the authors’ rebuttal has somewhat addressed these points, indicated by the reviewers' raising their scores to marginal acceptance. The results seem to show that the approach is applicable across multiple architectures and can enable both local and global concept-level interventions. Given these clarifications and the merits of the paper, I lean more toward acceptance and encourage the authors to revise the paper for the camera-ready version to address these concerns.

---

### Decision · Program_Chairs · 2025-02-11

Accept (Poster)